# A Miniature Fabry Perot Sensor for Twist/Rotation, Strain and Temperature Measurements Based on a Four-Core Fiber

**DOI:** 10.3390/s19071574

**Published:** 2019-04-01

**Authors:** Vedran Budinski, Denis Donlagic

**Affiliations:** Laboratory for Electro-Optic Sensor Systems, UM FERI, Koroška cesta 46, 2000 Maribor, Slovenia; denis.donlagic@um.si

**Keywords:** optical fiber sensor, rotation sensor, strain sensor, temperature sensor, multi core fiber, Fabry-Perot

## Abstract

In this article, a novel miniature Fabry-Perot twist/rotation sensor using a four core fiber and quadruple interferometer setup is presented and demonstrated. Detailed sensor modeling, analytical evaluation and test measurement assessment were conducted in this contribution. The sensor structure comprises a single lead-in multicore fiber, which has four eccentrically positioned cores, a special asymmetrical microstructure, and an inline semi-reflective mirror, all packed in a glass capillary housing. A four core fiber positioned in front of a special asymmetrical microstructure and the inline semi reflective mirror defines four Fabry-Perot interferometers. Rotation of the sensors’ asymmetrical microstructure around the axis of the in-line four core fibers´ modulates the path lengths of all four interferometers simultaneously. Proper processing of path length changes of all four interferometers allows for unambiguous and temperature independent determination of the sensor’s rotation angle.

## 1. Introduction

Over the last few decades, fiber optic sensors have become widely used in different applications, due to their distinctive and well known advantages, such as lightweight design, small size, immunity to electromagnetic interference (EMI) and high sensitivity. Fiber optic sensors have a reputation for high efficiency sensing of numerous physical parameters, such as strain, temperature and refractive index. Measurement of twist/rotation is also a basic and essential parameter measured in several industrial fields, such as industrial, geophysics, navigation, aeronautic, construction and the military systems. Hence fiber optic-based twist/rotation sensors have recently been investigated intensively. There have been several investigated approaches that depend either on measurements of circular birefringence [1,2,3,4] or linear birefringence [5,6,7,8,9,10,11,12,13], changes in twisted fibers, systems that exploit different effects related to Fiber Bragg Gratings (FBGs) [14,15,16,17], tilted FBGs [18,19,20,21,22] and Long Period Grating (LPGs) [23,24,25,26,27,28], E-field vector displacement in circularly symmetric fibers [29,30,31,32], multimode interference [33], and other systems based on specialty fibers and waveguides [34]. Many of the described approaches, however, suffer from limitations that can limit their functional usage. The majority of the sensors employ in-line configurations which require access to the sensor from both (opposite) directions, thus increasing the entire sensor setup to impractical proportions. Other configurations depend on specialized gratings, which are proved to introduce costly spectral interrogation techniques [35]. On the other hand, Fabry-Perot (FP) interferometry is presenting itself as a simple and compact alternative solution attempting to overcome the stated limitations. Several sensor systems have already been reported exploiting FP interferometry for measuring pressure [36,37], strain [38], temperature [39], and other physical parameters [40,41]. The greatest challenge when trying to design and fabricate a Fabry-Perot twist/rotation sensor is the modulation of the optical path length change of the FP interferometer instigated by twist/rotation. In current FP interferometer sensors applied pressure, strain, temperature, etc. produce linear displacements. On the other hand, in a homogeneous medium, i.e., glass or air, twist/rotation causes no variation of the optical path length of the FP interferometer. This paper presents a compact twist/rotation Fabry Perot (FP) sensor based on a four-core fiber. The sensor introduces an asymmetrical microstructure that modulates path length with respect to the twist/rotation of the sensor structure.

## 2. Sensor System Description

### 2.1. Sensor Design and Fabrication

The proposed sensor assembly is presented in Figure 1, and consists of a lead-in four core fiber, an asymmetrical microstructure positioned in front of the lead-in four core fiber, an in-line semi- reflective mirror and a support fiber, all packed in a thick wall glass capillary. The asymmetrical microstructure comprises of a short section of a coreless fiber, which is polished at an angle on one side, and spliced to the in-line fiber mirror and the support fiber on the other. The length of the support fiber is arbitrary, and serves primarily as a fastening element for attaching the sensor assembly to the measured object. The cores of the four core fiber are positioned tetragonally within a fiber cross-section plane, with distances between diagonally positioned cores of 70.7 μm (Figure 2a). The sensor assembly thus defines four FP interferometers, which are formed by the perpendicularly cleaved end of the lead-in four core fiber and the in-line fiber mirror, positioned between the asymmetrical microstructure and the support fiber. The optical path length of each FP interferometer is comprised of two paths: A path in the air and a path in the glass. The lengths of those two paths are, however, different for each individual interferometer, and, further, depend on the rotational alignment between the lead in four core fiber and asymmetrical microstructure, as shown in Figure 1. The angular displacement between lead-in fiber and support fiber, (which is permanently attached to the asymmetrical microstructure), thus modulates the total path lengths of all four interferometers in distinctive ways, as explained further below. 

The asymmetrical microstructure and the supporting fiber were fabricated from a coreless fiber. The sensor housing was made of a thick wall glass capillary with an inner diameter of 127.5 μm and outer diameter of 180 μm. The sensor production started by chemical etching of the coreless fiber with an initial diameter of 140 μm to a diameter of 127 μm, which corresponded closely (within tolerance better than a μm) to the measured diameter of the in-line four core fiber. To maximize the interferometer’s fringe visibility, the reflectance of the in-line mirror was adjusted to approximately 15%. This value was determined heuristically, and depended on signal loss caused by beam divergence within the open-path FP-cavity, and Fresnel loss, instigated by light passing through a tilted surface (asymmetrical microstructure) having different refractive indices (the tilted surface deflects the incoming and back reflected beam from the senor assembly axis, but within given geometrical parameters and beam divergence at the output of the four core fiber, reasonably good interference fringes were obtained for all interferometers). The in-line mirror was generated by sputtering a multi-layer film of TiO_2_ and SiO_2_ on the cleaved tip of a coreless fiber, which was then spliced to the support fiber by a fusion splicer by an appropriate splicing procedure to yield the preferred reflectivity [42]. The next manufacturing step included cleaving of the coreless fiber on one end, around 50 μm from the in-line mirror, and polishing it at an angle of approximately 15°, thus finalizing the fabrication of the asymmetrical microstructure. To achieve a respectable compromise between mitigating the sensors’ temperature influence on elongation (strain) measurements and the capability to measure temperature, the length of the asymmetrical microstructure was a few µm long [39,43]. Therefore, the angle polishing process ended when the shorter side of the asymmetrical microstructure was in the vicinity of the in-line mirror (Figure 2b). The final step included fitting the in-line four core fiber and the asymmetrical microstructure into the thick wall glass capillary housing. To assure proper fixation of the thick glass capillary housing, we attached it firmly to the lead-in four core fiber with a high Young modulus epoxy glue, to prevent any torsional creep of the fiber. The inner wall of the glass capillary was smooth and without irregularities in the change of the inner diameter, so it did not cause noticeable influence on the measurements. The distance between the in-line four core fiber and the asymmetrical microstructure with the supporting fiber depended on a sufficient fringe visibility of all four cores (interferometers), and was determined practically by monitoring them with the interrogation system. 

### 2.2. Sensor Operation

As already described in the previous section, the perpendicularly cleaved end of the in-line four core fiber (M_1_) and the in-line mirror (M_2_) define four FP interferometers. We assume that when no twist/rotation is applied to the sensor, core c_1_ is positioned against the shorter side of the asymmetrical microstructure h_1_, thus the light wave within the resonator trespasses the longest distance in the medium with the refractive index n_1,_ and the shortest distance in the medium with the refractive index n_2_. Diametrically located core c_3_ is positioned against the longer side of the asymmetrical microstructure h_2_, therefore, the light wave travels the shortest distance in the medium with the refractive index n_1,_ and the longest distance in the medium with the refractive index n_2_. The optical path lengths of interferometers defined by cores c_2_ and c_4_ are the same in this initial position. Exposing the sensor to twist/rotation alters the position of cores with respect to the angled plane of the asymmetrical microstructure. Rotation thus causes variation of optical path length in all four interferometers. For example, when the sensor is subjected to a 180-degree twist/rotation, diametrically positioned core c_1_ and core c_3_ interchange their positions, i.e., core c_3_ becomes positioned against the shorter side of the truncated cylinder h_1,_ and core c_1_ is sited against the longer side of the truncated cylinder h_2_. Thus, the optical path length of the FP interferometer defined by the core c_1_ becomes longer than the optical path length of the interferometer defined by c_3_ (Figure 3b).

When the sensor is subjected to twist/rotation of an angle Φ from the initial state, the optical path lengths of all four interferometers change, as the ratio between path lengths of the light waves traveling in the medium with refractive index n_1_ and in the medium with the refractive index n_2_ change. Accordingly (and using Figure 3 and Figure 4), the path length of the light wave traveling in the medium with refractive index n_1_ can be expressed as:(1)$$l_{n1,N}=l^{\prime }+\frac{d}{2}\tan\delta_{p}\sin\left(\Phi +\left(N-1\right)\frac{\pi }{2}\right),\;\;N=1,2,3,4,$$
where *l′* represents the initial path length of the light wave in the medium with refractive index n_1_, *d* represents the distance between diametrically positioned cores, *δ_p_* represents the angle between the longitudinal axis and polished plane of the asymmetrical microstructure, while *N* denotes the index of cores of the four core fiber. Furthermore, the path length of each individual interferometer in the medium with refractive index n_2_ can be expressed as:(2)$$l_{n2,N}=l+l^{\prime }-l_{n1,N}.$$
where *l* represents the initial path length of the light wave in the medium with refractive index *n*_2_, *l_n_*_1,*N*_ represents the optical path length in the medium with refractive index *n*_1_. Thus, the optical path length of the sensor’s FP interferometer, defined by the core with index *N,* can be expressed as:(3)$$OPL_{N}=l_{n1,N}\cdot n_{1}+l_{n2,N}\cdot n_{2}\to OPL_{N}=l_{n1,N}\left(n_{1}-n_{2}\right)+\left(l+l^{\prime }\right)n_{2}$$

When the sensor is subjected to twist/rotation by angle *Φ*, optical path length change can be expressed as:(4)$$\Delta OPL_{N}=OPL_{N,0}-OPL_{N,\Phi }.$$

Considering Equations (1), (3) and (4), the above expression for optical path length change of each FP-Interferometer (N = 1, 2, 3, 4) can be expressed as:(5)$$\Delta OPL_{N}=\frac{d}{2}\tan\delta_{p}\left(n_{1}-n_{2}\right)\left(\sin\alpha_{N,\Phi }-\sin\alpha_{N,0}\right)+\left(l_{\Phi }^{\prime }-l_{0}^{\prime }\right)n_{1},\;\;N=1,2,3,4.$$

The optical path length of individual interferometers depends on the distance between fibers in the center of the structure l’, which also depends on strain and temperature. To obtain strain and temperature insensitive information on angle Φ, optical path length changes from two diametrically positioned FP-Interferometers (i.e., N = 1 and N = 3 or N = 2 and N = 4) can be further subtracted. By using Equations (1) and (5), twist/rotation angle from optical path length changes in FP-Interferometers N = 1 and N = 3 can be expressed as:(6)$$\sin\Phi =-\frac{\Delta OPL_{3}-\Delta OPL_{1}}{d\tan\delta_{p}\left(n_{1}-n_{2}\right)}+\sin\Phi_{0}.$$

Angle Φ_0_ in Equation (6) is determined by the initial core alignment with respect to the asymmetrical microstructures’ plane angle (Figure 4). When Φ_0_ = 0, diametrically positioned FP interferometers N = 1 and N = 3 are located in the center of the asymmetrical microstructure slope, thus, their optical path lengths are identical (dashed region (c) on Figure 5a). The optical path length of FP interferometer N = 2 is the shortest (dashed region (b) on Figure 5), while the optical path length of its diametrically positioned FP interferometer N = 4, is the longest (dashed region (a) on Figure 5a). 

By subtracting path length variations of diametrically positioned FP interferometers, i.e., N = 1 – N = 3 and N = 2 – N = 4, two output signals are obtained, which are shifted in phase by 90 degrees and possess no offset value, and, thus, allow for unambiguous rotation angle reconstruction over the entire 2π rotation range (Figure 5b).

The proposed sensor also enables measurements of elongation (strain) and temperature, as strain and temperature modulate the distance between the semireflective mirrors and the refractive index of the glass respectively. Strain and/or temperature variations modulate the optical path lengths of all four FP interferometers simultaneously and proportionally. To distinguish between strain or temperature effects, appropriate operations shall be implemented among the acquired optical path length changes for all four FP interferometers. For example, elongation ΔL can be extracted by calculating average change in length of all four FP interferometers:(7)$$\Delta L=\frac{\left(\Delta OPL_{1}+\Delta OPL_{3}\right)+\left(\Delta OPL_{2}+\Delta OPL_{4}\right)}{4}.$$

The sum of optical path length changes of diametrically positioned FP interferometers shall be subtracted to extract the temperature:(8)$$\Delta T=\frac{\left(\Delta OPL_{1}+\Delta OPL_{3}\right)-\left(\Delta OPL_{2}+\Delta OPL_{4}\right)}{\frac{dn_{SiO_{2}}}{dT}L_{SiO_{2}}}.$$
where L_SiO2_ corresponds to the path length of interferometers in the asymmetrical microstructure, and dn_SiO2_/dT represents the change of fiber refractive index due to temperature change.

### 2.3. Sensor Interrogation

Measurements of the optical path length changes for each FP interferometer were obtained by phase-tracking of characteristic components with Inverse Discrete Fourier Transform (IDFT) of the sensor’s Back-Reflected Optical Spectrum (BROS) [44,45,46]. In our experimental characterization, we used a NI PXIe-4844 spectral interrogator (National Instruments) to acquire sensors’ spectral characteristics (we performed 80 nm wide sweeps with a 10 Hz repetition rate). Prior to the IDFT, spectral data were converted from the optical wavelength domain into the optical frequency domain. The complex IDFT data contain local peaks in their absolute values, which correspond to round trip time-of-flights of individual FP interferometers (Figure 6). Phases of these complex peak values (φ = arctan(Im/Re)) further correspond to the relative positions of spectral components (spectral fringes) of individual FP interferometers in the optical frequency spectrum [46]. Thus, by calculating the phases of complex values that otherwise correspond to the local absolute peaks in IDFT data, one can track small changes in an individual interferometer’s length changes reliably:(9)$$\Delta OPL_{N}=\frac{\lambda }{4\pi }\Delta \varphi_{N}.$$

To calculate twist/rotation angle, elongation and temperature according to Equations (7)–(9) respectively, data from all four FP interferometers’ phases were recorded with the spectral interrogator.

## 3. Experimental Results

The experimental setup is illustrated in Figure 7. A special fan-out unit (FAN-4C, Fibercore) was used to couple light between the in-line four core fiber and four separate single core fibers, which were then connected to a four channel NI PXIe-4844 spectral interrogator. The sensor was fixed between two clamps. For measuring twist/rotation, a second clamp was mounted on a rotational stage with an angular resolution of 0.01°. For measuring elongation, the same clamp was attached to a linear stage with a resolution of 1 μm.

The rotational stage was used to impose rotation/torsional twist to the sensor, as shown in Figure 7. The sensor was rotated in a clockwise, and afterwards in a counter clockwise direction, by a mechanical angle of 90° in 10-degree steps. Figure 8 shows a normalized phase readout from diametrically opposite FPIs’ (N = 1 and N = 3) and calculated rotational angle using Equation (6).

To estimate the viable resolution of the system, the rotational stage was moved periodically in clockwise and counter clockwise directions while reducing the amplitude of these movements. Figure 9 shows rotation in clockwise and counter-clockwise directions by 0.02 degrees, which is well above the measurement noise. The measurement data were averaged five times by the measurement algorithm, i.e., the sampling rate of the spectral interrogator was decreased to 5 Hz. From Figure 9, an angular resolution of about 0.02 degrees is achievable with the presented setup.

Furthermore, the sensor’s response was measured to the change of length (elongation). By “stretching” the sensor by 4 μm at 1 μm per step, and back to the initial position (Figure 10), the elongation was calculated by Equation (7).

Parallel with elongation measurement, the influence of the latter on the twist/rotation angle was evaluated (simply by pressing the data in a way to calculate the rotation angle). In this test, we applied a constant temperature of roughly 25 °C to the sensor, and the rotation angle was set to around 0 degrees. Results indicate changes in the calculated twist/rotation angle of about 0.013° (Figure 11a). Possible grounds for the inconsistency in the twist/rotation angle readout could arise from the vibrations of the motors, built in the linear stages, which might have altered the uneven change in the optical path length of the diametrically opposite FP interferometers. Furthermore, we also investigated the possible temperature impact on the elongation measurements (Figure 11a). We calculated the temperature with Equation (8). Since the length of the temperature sensitive part of the sensor, i.e., the asymmetrical microstructure made out of silica was few µm long, the recorded temperature influence did not exceed 0.65 °C.

Finally, we recorded the sensor’s response to the temperature change. For testing objectives, we designed a special tubular shaped temperature chamber. Temperature was calculated by Equation (8). The sensor was tested in a temperature range between 25 °C and 125 °C (Figure 12). For reference measurements we used a PT100 temperature sensor.

Figure 12 compares the actual temperature change with the proposed sensor’s response over time (the reference sensor was a PT100). Furthermore, we monitored the calculated twist/rotation angle change (Figure 13), which evolved during the same temperature cycle.

As depicted in Figure 13, the largest deviation in the measured twist/rotation angle was not excited i.e., 0.012 degrees. The local peak in amplitude of this deviation coincided with the initial part of the temperature characteristic, which also has the highest temperature gradient (see Figure 12). Thus, this deviation could probably be attributed to the temperature gradients that developed across diametrically positioned FP interferometers (the drift would probably be even lower if the change in temperature were slower).

## 4. Conclusions

This paper presented a miniature Fabry Perot twist/rotation sensor with a special asymmetrical microstructure, which allows modulation of the optical path length when the sensor is subjected to twist/rotation. The asymmetrical microstructure was fabricated by angle polishing of the coreless fiber. The perpendicularly cleaved end of the in-line four core fiber and the in-line mirror define four FP interferometers with rotation-dependent path lengths. The measured ambiguous range of the sensor was ±90°. The sensor was also tested for elongation and temperature measurements. The influence of strain and temperature on rotation sensing proved to be below 0.015°, even over relatively wide strain and temperature spans.

## Figures and Tables

**Figure 1 sensors-19-01574-f001:**
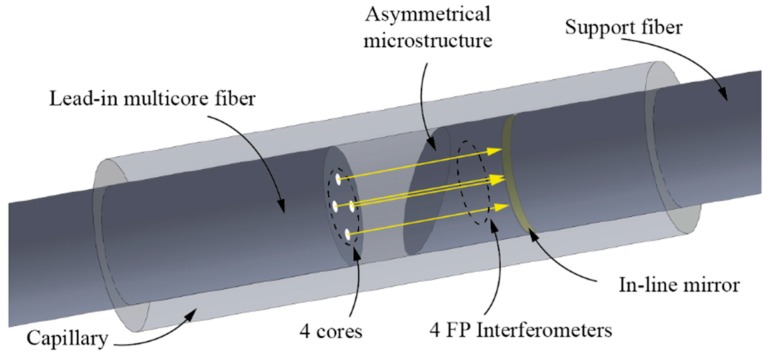
Sensor design.

**Figure 2 sensors-19-01574-f002:**
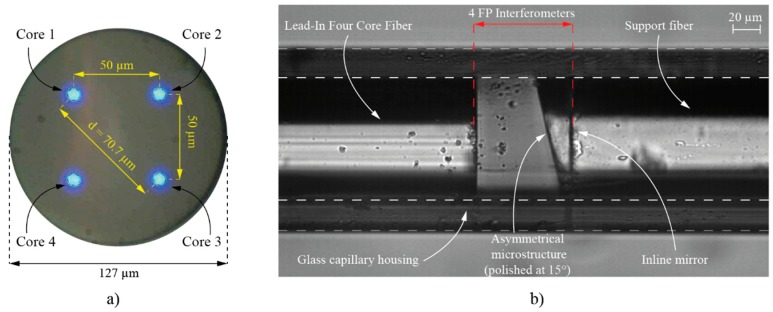
(**a**) Cross-section view of the four-core fiber and (**b**) Sensor assembly.

**Figure 3 sensors-19-01574-f003:**
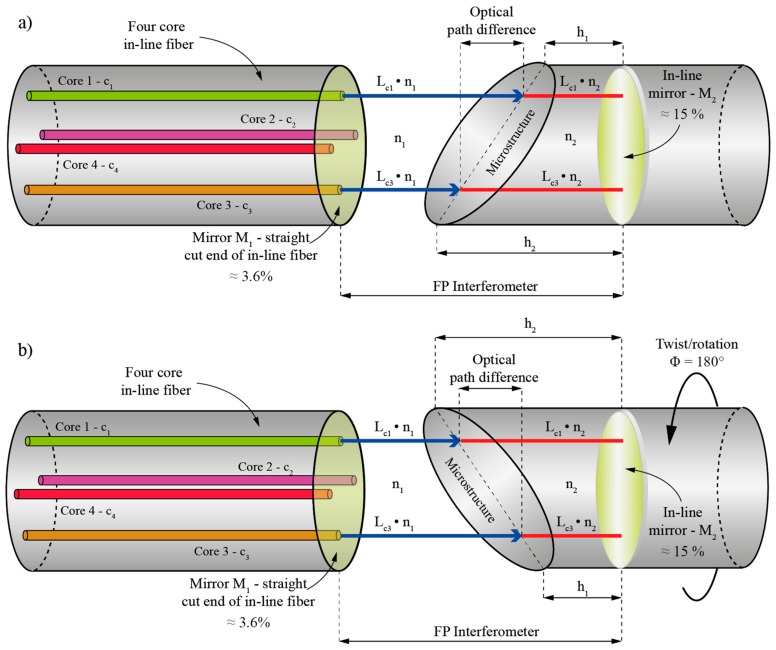
(**a**) Relaxed sensor and (**b**) 180-degree twist/rotation applied to the sensor.

**Figure 4 sensors-19-01574-f004:**
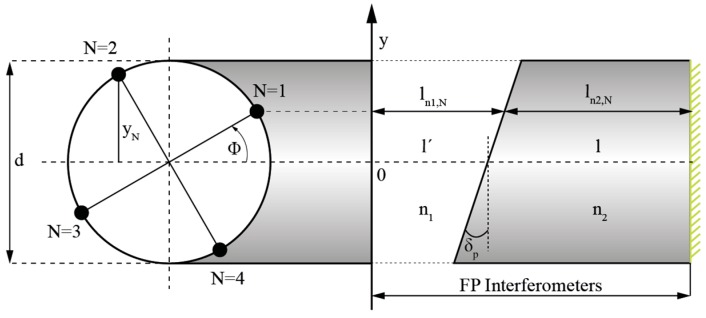
Principle of operation.

**Figure 5 sensors-19-01574-f005:**
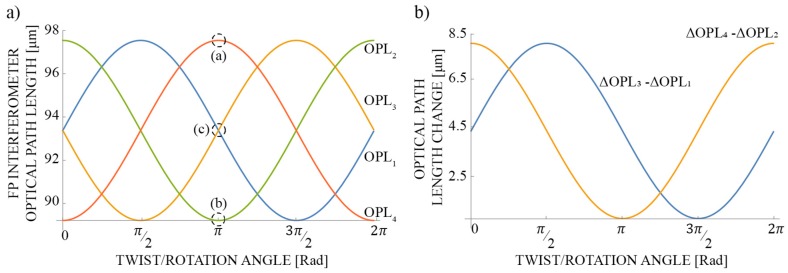
(**a**) Optical path length change of all FP-Interferometers (from Equation (5)) when the sensor is rotated by a “mechanical” angle 0 ≤ Φ ≤ 2π. Physical parameters for the sensor in the simulation: d = 70.7 µm, δ_p_ = 15°, n_1_ = 1, n_2_ = 1.444 and (**b**) Subtracted path length variations of diametrically positioned FP-Interferometers (N = 1 – N = 3 and N = 2 – N = 4).

**Figure 6 sensors-19-01574-f006:**
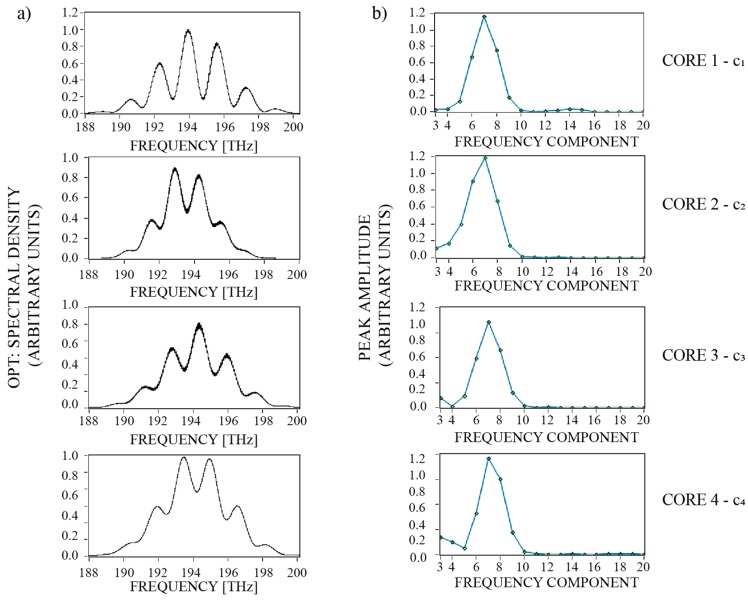
(**a**) Typical recorded optical back-reflected spectrum of FP interferometers (with applied Gaussian window) and (**b**) their amplitude (Fourier transform).

**Figure 7 sensors-19-01574-f007:**
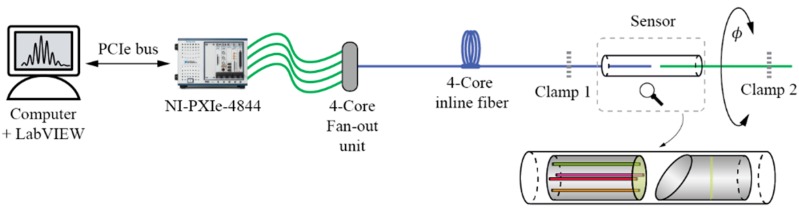
Experimental setup.

**Figure 8 sensors-19-01574-f008:**
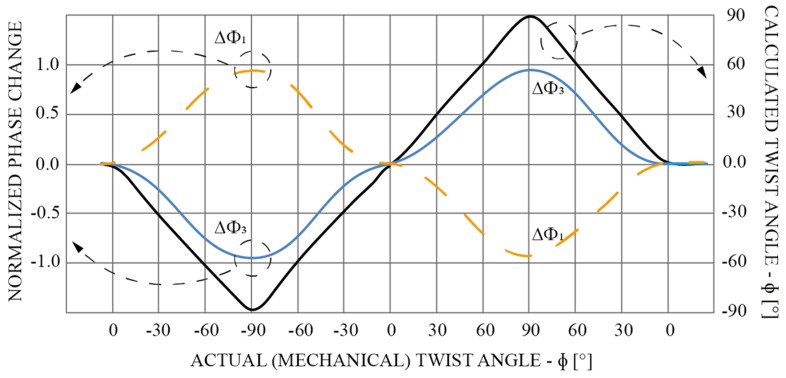
Calculated twist/rotation angle from diametrically opposite FPIs’ phases measurement. The dashed yellow curve and blue curve represent normalized phase change of diametrically opposite FPIs, and the black curve represents the calculated twist angle.

**Figure 9 sensors-19-01574-f009:**
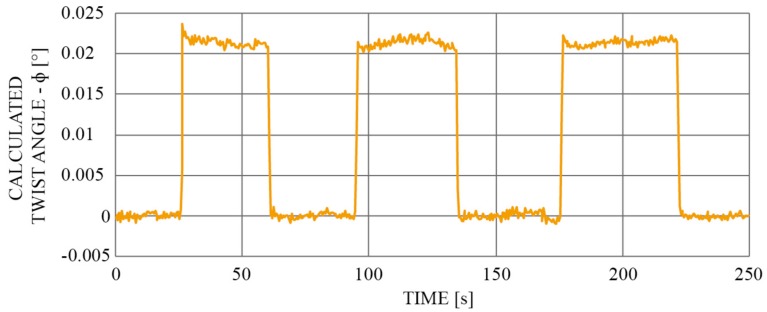
Demonstration of the sensor’s sensitivity: Left and right movement of the rotational stage by 0.02 degrees (mechanical).

**Figure 10 sensors-19-01574-f010:**
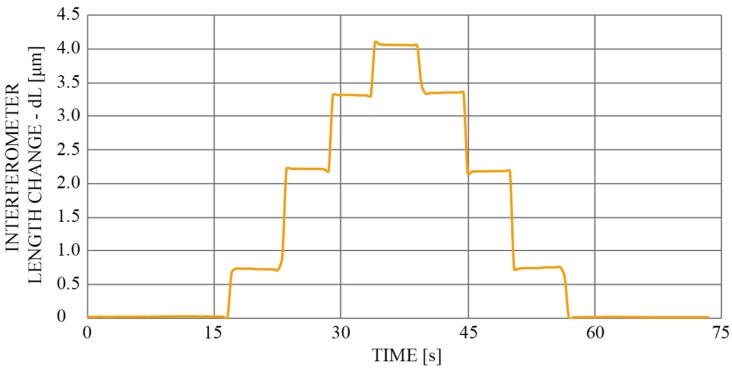
Calculated sensor elongation for 4 µm.

**Figure 11 sensors-19-01574-f011:**
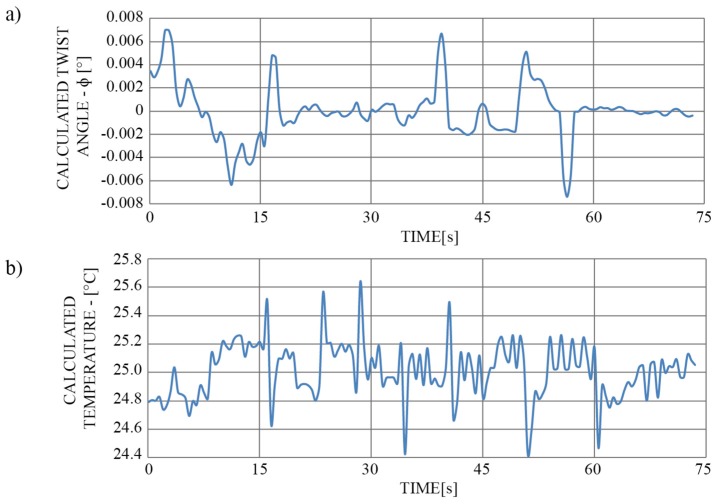
(**a**) Influence of the change in the sensor length for 4 µm on the twist/rotation angle readout and (**b**) Calculated temperature change instigated from elongation.

**Figure 12 sensors-19-01574-f012:**
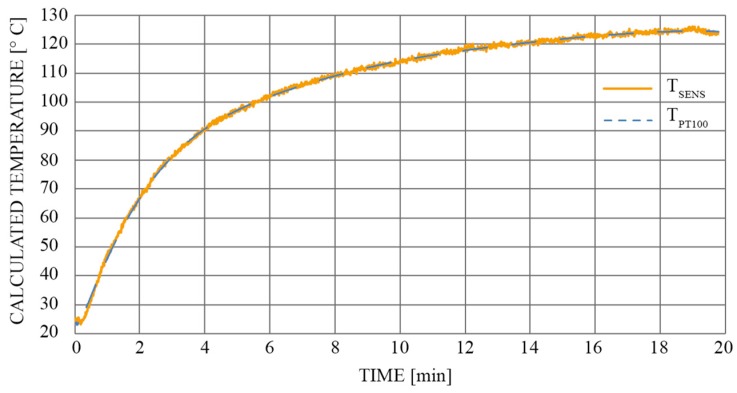
Temperature measurement and comparison with a reference temperature sensor (PT100).

**Figure 13 sensors-19-01574-f013:**
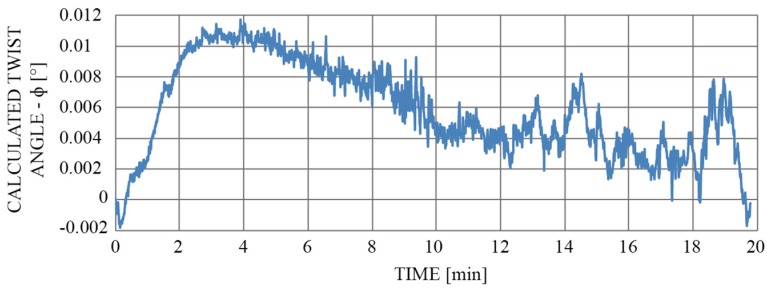
Influence of the change in temperature on the calculated twist/rotation angle measurement when the sensor was heated by about 100 °C.

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
