# Peer review of "A Miniature Fabry Perot Sensor for Twist/Rotation, Strain and Temperature Measurements Based on a Four-Core Fiber"

_sensors, 2019, doi:10.3390/s19071574_

Reviewer 1 Report

The authors present the sensor for simultaneous measurement of multi parameters base on Fabry-Perot interferometers. The sensor consists of four core fiber, small air gap, as authors called asymmetrical microstructure and mirror for reflection. According to my opinion the proposed structure of sensor is very interresting. I see the limitations in using of the special interogation system.

Nevertheless, the asymmetrical microstructure has dimmension of micrometers, the name is quite confusing. I was looking some microstructure as at microstructure fiber. And it is only angled end face no-core fiber.

Minnor corrections:

row 27 FOS is not introduced

31 long period grating LPG

33 capital letter Current

84 four insted of fur

94 later?

94 What distance was suitable for suitable visibility? How do you check it?

125 p should be as index

145 angle instead of angel

228 the instead of de

Author Response

Dear Reviewer,

please find our response in the attached document (pdf).

Best regards,

Authors

Reviewer 2 Report

 A twist sensor based on a FPI was proposed by the authors. The sensor comprises of a four-core fiber, a 15° polished fiber and an in-line mirror in order to measure the twist angle through optical path difference when the structure is twisted. In general, the work is interesting and well presented. Thus, the paper should become acceptable after some corrections/clarifications listed below:

- Some misspellings throughout the manuscript such as "if" (in line 68), "angle" (line 145) and "de" (line 228). Authors should check the manuscript for similar errors;

- I suggest the authors add some words about the capillary shown in Figure 1. How is it attached to the fibers? What is its influence on the sensor response? Is it twisted with the fibers or fixed and the fibers twist inside the capillary?

- Please, add a legend or caption in Figure 8 in order to indicate what each curve means.

- Please, clarify for the readers how Figure 11 was obtained. Is it a stabilization test (did you applied a constant angle/temperature and see how the sensor deviates from the applied angle/temperature)? 

- The authors claim in the conclusion section (line 264-266) that it is possible to simultaneously measure strain, twist and temperature. However, the authors did not show this feature on the manuscript and not mentioned how it can be possible to measure this parameter simultaneously. If you are only monitoring the phase change with each parameter, how is that possible to isolate strain, temperature and twist in the case which all these variables are varying simultaneously? 

Author Response

(The authors gave the same response as above.)

Reviewer 3 Report

The authors present a very interesting work regarding the use of 4 core fiber for rotation and elongation monitoring. Nevertheless, the work here presented is very similar to the one already reported by the same authors in the conference proceedings paper:  

Budinski, Vedran, and Denis Donlagic. "Miniature Twist/Rotation Fabry Perot Sensor Based on a Four-Core Fiber." In Multidisciplinary Digital Publishing Institute Proceedings, vol. 2, no. 13, p. 1091. 2018.

It is the reviewer understanding, that no considerable new results were added to the manuscript under review. Actually most of the figures are exactly the same.

Although the reviewer would agree with the value of the work, it clearly needs new results/information/inputs to be differentiated from the previous conference paper.

Author Response

Dear Reviewer,

please find our response in the attached document (pdf).

Best regards,

Authors

Round  2

Reviewer 3 Report

The reviewer opinion is that the work presented by the authors is indeed valuable, nevertheless, was already published in the conference paper:

Budinski, Vedran, and Denis Donlagic. "Miniature Twist/Rotation Fabry Perot Sensor Based on a Four-Core Fiber." In Multidisciplinary Digital Publishing Institute Proceedings, vol. 2, no. 13, p. 1091. 2018.

It was NEVER the reviewer suggestion for the authors “just to change the look of the existing figures for the purpose of this paper”. That would be indeed UNETHICAL.

The reviewer suggestion was for the authors to complement the data already existing in the paper in order to add information not published yet. That request was not considered in the authors revision, and therefore the reviewer cannot change recommendation previously given.

Author Response

Dear Reviewer,

please find our comments and corrections in the attached document.

Best regards,

Authors
